# Simulation Test on Cooling and Fire Suppression with Liquid Nitrogen in Computer Room of Data Center

**Jianbing Meng [1], Tingrong Wang [2], Guanghua Li [2] and Jianhong Kang [2,\*]**

[1] School of Safety Science and Engineering, Anhui University of Science & Technology, Huainan 232001, China
[2] Jiangsu Key Laboratory of Fire Safety in Urban Underground Space, China University of Mining and Technology, Xuzhou 221116, China
\* Correspondence: jhkang@cumt.edu.cn

**Abstract:** With the rapid development of worldwide computer data center construction, the reliability requirements of the fire-fighting system for data center rooms are also increasing. By using the self-designed simulation platform of liquid nitrogen spray, this paper studies the liquid nitrogen cooling process in the initial heating stage of a computer data center room fire and the liquid nitrogen extinguishing effects for common combustible materials, revealing the feasibility of applying liquid nitrogen to the fire extinguishing system for data center room. The results show that the cooling and inerting effects with quarter sector fan-shaped 6520 spray nozzle are the best among seven types of spray nozzles, the relative temperature changes by more than 50% within 5 min, and the oxygen concentration in the test space drops below 10%. Compared with optical fiber, the ignition range of uninterruptible power supply com-bination during combustion is relatively small. Liquid nitrogen has a significant fire-extinguishing effect on two combustible materials, which can successfully extinguish optical fiber and UPS within 3 min and 2 min, respectively.

**Keywords:** liquid nitrogen; data center; fire suppression; cooling

## 1. Introduction

With the rapid development of the Internet and information technology, the world has entered the DT era from the IT era. The global construction of super-large data centers has increased from 259 in 2015 to 597 in 2020, and is still increasing [1,2]. However, since a large number of electronic devices in the data center room run continuously, the equipment is prone to excessive load and unreasonable circuit design, leading to security risks [3–5]. In 2009, a transformer fire at Fisher Plaza data center in Seattle caused a fire. In 2008, a data center in Wisconsin caught fire due to aging cables. In 2015, a cable fire broke out at the Delta Telecom Data Center in Azerbaijan. Data center fires will not only cause loss of life and property, but even affect the normal operation of the whole society [6,7]. Therefore, a safe and reliable fire protection system is very important for the stable operation of data center room.

In order to reduce the occurrence of fire in data center machine room, scholars have done a lot of research on heat flow distribution and fire prevention system in data center machine rooms. Brian [8] analyzed the effects of fuel type, compartment configuration, environmental conditions, and maximum allowable size on electrical equipment and cable fires, and found that smoke growth rate, smoke characteristics, ambient temperature, induced airflow of the ventilation system, and detector response characteristics had an impact on early fire detection. Sorell et al. [9] simulated the influence of different ceiling heights on gas flow in the computer room, and concluded that the ceiling could lead to blocked return air, slow down heat dissipation of the body and more easily cause fire. Bhagwat [10] et al. studied the cooling effect of the machine room by using the heat impact index and proposed an optimization scheme Volk et al. [11] evaluated the overall cooling

effect of the data center by collecting the cold and hot air flow data of the data center room. Murray et al. [12] applied artificial neural network to study the gas flow and temperature distribution in the data center room. Marco et al. [13] found that noise emitted by nozzles used in data center machine rooms would lead to performance attenuation of hard disk drives. By comparing various nozzles under several operating conditions, it was concluded that under the same mass flow rate released, the higher the fluid density, the lower the noise generated. Zeng et al. [14] used medium scale parallel plate fire test (PPT), small scale fire transmission device (FPA) test and smoke corrosion test data center cables to show that low-smoke halogen-free cable has good fire and smoke corrosion performance. Nada et al. [15] studied the floor opening rate of data center machine room and analyzed the influence of the opening rate on the flow of cold and hot air in the machine room. The design specification of the computer room in China has strict requirements on the selection of fire extinguishing system, including gas fire extinguishing system (HFc-227ea, IG541, and aerosol fire extinguishing agent) and water matrix fire extinguishing system (preaction and high-pressure fine water mist, etc.) [16–18]. However, all kinds of gas extinguishing agents have some defects, including high price, limited use, etc. Water matrix fire extinguishing has greater damage to the performance of equipment in machine room, so the search for new fire extinguishing materials is the current urgent task. In recent years, liquid nitrogen, as a safe, reliable, environmentally friendly and low-cost fire extinguishing agent, has attracted many scholars' attention [19,20]. Liquid nitrogen can decrease the oxygen concentration in the fire space (inert effect), and the vaporization process of liquid nitrogen absorbs a lot of heat from the outside, so that the surface temperature of the burning object drops rapidly below the ignition point (cooling effect) [21]. Levendis [22] introduced the application technology of a remote-controlled drone carrying liquid nitrogen to extinguish pool fires. Zhou et al. [23] invented a new technology for underground coal fire suppression by direct infusion of liquid nitrogen. Shi et al. [24] studied the influence of liquid nitrogen flow rate and injection distance on open oil pool fire. Some scholars also studied the application of liquid nitrogen in fire suppression in the goaf of coal mines [25,26].

Liquid nitrogen fire extinguishing system is not only safe and environmentally friendly, but also has no damage to sensitive electronic components, so it can be used as an ideal fire extinguishing agent in data center rooms. At present, there is little research on the application of a liquid nitrogen system in the fire suppression of data center room, and the feasibility of fire suppression with liquid nitrogen for data center is not clear. To this end, this paper uses a self-built liquid nitrogen cooling and fire extinguishing test platform to analyze the cooling and inert effects of liquid nitrogen on the data center room fire at different nozzle flows, and studies the fire extinguishing effects of liquid nitrogen with two common combustible materials in the data center. The feasibility of fire suppression with liquid nitrogen in data center room was preliminarily discussed.

## 2. Materials and Methods

### 2.1. Simulation Test System

The simulation of fire suppression with liquid nitrogen in computer data center room includes two parts: cooling simulation and extinguishing simulation. A cubic container with a side length of 1m and made of acrylic is used to simulate the data center room, and six boxes dimensioning the length, width and height of 25 cm × 25 cm × 35 cm placed inside the container are used to simulate the computers in data center room. Froude scaling criterion is employed to build the connection between the full-scale and the reduced-scale experimental environment [27,28]. The scaling relationships of variables under various scales can be simplified: $Q_M/Q_F = (l_M/l_F)^{5/2}$, $T_M = T_F$, $T_M = T_F$. Where $Q$ is the heat release rate of the fire source, kW; The subscript '$M$' and '$F$' represent the model and full-scale parameters, respectively. $l_M/l_F$ is the similarity ratio of geometric dimensions. The test system is mainly composed of a self-pressurized liquid nitrogen tank, freeze-proofing electronic balance, nozzle, temperature sensor and thermostatic PTC ceramic heater, as shown in Figure 1a. The 4OXV oxygen sensors and heating sheets are arranged in the

container. One side of the container has a liquid nitrogen injection hole which is connected with the liquid nitrogen tank through the stainless steel pipe.

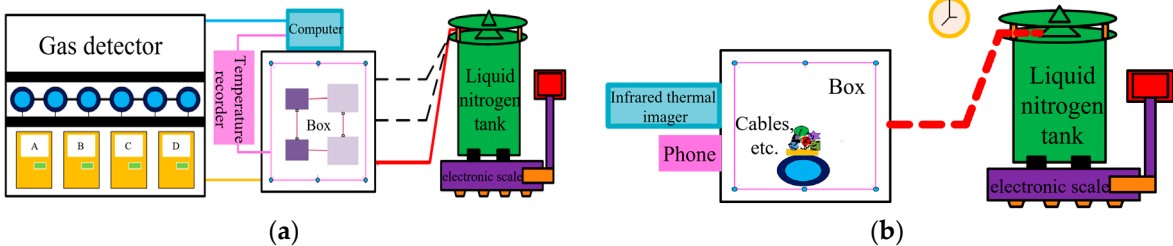

**Figure 1.** Diagram of test system: (**a**) Liquid nitrogen cooling test system; (**b**) Liquid nitrogen fire extinguishing test system.

The test platform can also simulate the fire extinguishing effect of liquid nitrogen for different combustible materials. Additional equipment such as liquefied butane gas cylinder (igniter) is needed, as shown in Figure 1b. The test bench has an objective table, which can place different combustible materials.

### 2.2. Simulation Test Scheme

### 2.2.1. Test Content

(1) Liquid nitrogen cooling simulation. The whole test table is heated by heating sheets to simulate the temperature rise process in the early stage of fire. Under the operating pressure of 0.2MPa, the low temperature liquid nitrogen enters the container through the stainless steel pipe at a certain flow rate. The effects of liquid nitrogen injection on the temperature change and oxygen concentration distribution in different positions of the container are obtained.

(2) The simulation of fire suppression with liquid nitrogen. Fire hazards in data centers mainly include electric cable, network cable, power supply cord, uninterruptible power supply (UPS) batteries and other combustible materials. Due to the possibility of explosion for UPS battery combustion, for the sake of safety, the UPS control board and switching transformer are used to replace the UPS batteries. During the simulation test, the combustion materials such as optical fiber, cable and UPS are placed on the objective table, and the extinguishing effect of liquid nitrogen injection on different materials is simulated.

(3) Liquid nitrogen nozzle selection. Liquid injected from different nozzles produce different atomization areas, which will have a certain influence on liquid nitrogen cooling and fire extinguishing. In this test, quarter sector fan-shaped spray nozzles (6510#, 6520#, 6530#, 11010#, 11020#) and quarter solid conical spray nozzles (5B#, 10B#) are chosen. The physical picture and parameters are shown in Figure 2, Tables 1 and 2.

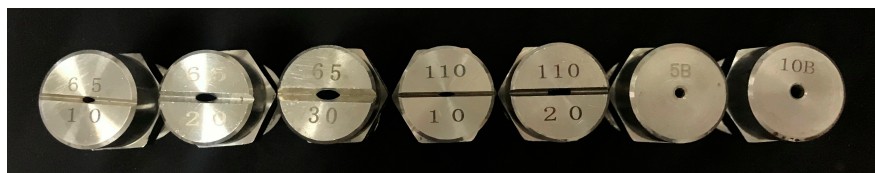

**Figure 2.** Physical map of different types of liquid nitrogen nozzles used in the test.

**Table 1.** Parameters of quarter sector fan-shaped spray nozzle.

| Type | Equivalent Diameter (mm) | Flow (L/min) | | | | | | | | | |
|------|---|---|---|---|---|---|---|---|---|---|---|
| | | **0.3 Bar** | **1 Bar** | **2 Bar** | **3 Bar** | **4 Bar** | **5 Bar** | **6 Bar** | **7 Bar** | **10 Bar** | **20 Bar** |
| 6510# | 2.0 | 1.0 | 2.3 | 3.2 | 3.9 | 4.6 | 5.1 | 5.6 | 6.0 | 7.2 | 10.2 |
| 6520# | 2.8 | 1.9 | 4.6 | 6.5 | 7.9 | 9.1 | 10.2 | 11.2 | 12.2 | 14.4 | 20 |
| 6530# | 3.6 | 2.5 | 6.8 | 9.7 | 11.8 | 13.7 | 15.3 | 16.7 | 18.1 | 22 | 31 |
| 11010# | 2.0 | 1.2 | 2.3 | 3.2 | 3.9 | 4.6 | 5.1 | 5.6 | 6.0 | 7.2 | 10.2 |
| 11020# | 2.8 | 2.5 | 4.6 | 6.5 | 7.9 | 9.1 | 10.2 | 11.2 | 12.1 | 14.4 | 20 |

**Table 2.** Parameters of quarter sector solid conical spray nozzle.

| Type | Equivalent Diameter (mm) | Flow (L/min) | | | | | | | | | |
|------|---|---|---|---|---|---|---|---|---|---|---|
| | | **0.5 Bar** | **0.7 Bar** | **1.5 Bar** | **2 Bar** | **3 Bar** | **4 Bar** | **5 Bar** | **6 Bar** | **7 Bar** | **10 Bar** |
| 5B# | 2.0 | 1.6 | 1.9 | 2.7 | 3.1 | 3.7 | 4.2 | 4.7 | 5.1 | 5.5 | 6.5 |
| 10B# | 3.18 | 3.3 | 3.8 | 5.4 | 6.2 | 7.4 | 8.5 | 9.4 | 10.2 | 11.0 | 13.0 |

2.2.2. Data Acquisition and Processing

The experimental data are acquired by PC processor, temperature acquisition device, and gas detection system. Temperature is measured by K-type thermocouple. Monitoring data of temperature and gas concentration acquired in real time are transmitted to the processing terminal by the TracerDAQ software. The data is exported by SmartView3.6 software, and then Matlab is used to extract the internal temperature data of the measured object for statistical analysis.

The liquid nitrogen injection hole with a diameter of 23 mm is located in the center of one side of the cubic container. A total of 12 4OXV oxygen sensors and 18 K-type thermocouples are placed inside the test container where eight concentration monitoring points are on the eight corners of the container and the other four are arranged on the four edges of the container. In addition, twelve temperature monitoring points are arranged at the same position as oxygen sensors, and the other six are placed to be close to the heating sheets on the six boxes. The arrangement of the monitoring points is shown in Figure 3.

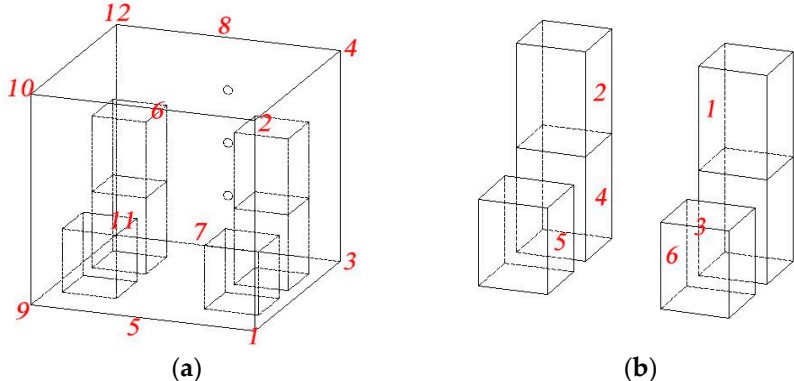

(**a**)                    (**b**)

**Figure 3.** Monitoring points of temperature and oxygen concentration sensors: (**a**) The sensors arrangement on the container; (**b**) Inside temperature monitoring points on the boxes.

*2.3. Test Procedure*

(1) Debug the oxygen sensor, temperature sensor and other equipment. Then, connect the stainless steel pipe and liquid nitrogen tank which is placed on the freeze-proofing electronic balance, and put the other end of the nozzle into the injection hole.

(2) Turn on the power switch. When the temperature of the heating sheets on the boxes rises to 70 °C, immediately turn on the liquid nitrogen tank to spray liquid nitrogen.

Stop spraying liquid nitrogen when the indication of electronic balance drops to the set value.

(3) Collect the data of temperature and gas sensors for 30 min, draw out the stainless steel pipe, and replace different types of nozzles after the temperature and oxygen concentration return to the initial environment.

(4) Open the container cover until the temperature and oxygen concentration recover the initial values, then replace another type of nozzles.

(5) Repeat steps 2–4 until every type of nozzles are tested.

(6) Open the container cover, place one kind of combustible material on the objective table, and debug the equipment.

(7) Ignite the combustible material and continue burning for 30 s.

(8) Open the switch of the liquid nitrogen tank until the flame of the combustible materials is completely extinguished, and record the value of the electronic balance.

(9) Open the container cover until the temperature and oxygen concentration recover the initial values, then replace another kind of combustible material.

(10) Repeat steps 6–9 until all kind of combustible materials are tested.

In the process of steps 2–3 and step 7–8, open the data acquisition system simultaneously.

## 3. Results and Discussion

When liquid nitrogen is used for cooling and extinguishing fire, the injected liquid nitrogen must occupy more than 33% of the space of the fire site. The amount of liquid nitrogen injected should be within a reasonable range because there are many electronic devices in the data center room and their ability to withstand low temperature is different. Through repeated attempts and considering various influencing factors, the mass of liquid nitrogen injected in this test is controlled to be less than 0.65 kg. The initial temperature in the computer data center room is about 20 °C, and the ambient pressure is 101,325 Pa.

### 3.1. The Variations of Temperature in Cooling Process

The quarter sector fan-shaped nozzles in the test include 65 series (6510#, 6520#, 6530#) and 110 series (11010#, 11020#). As shown in Figure 4, when 65 series nozzles are used, the monitoring positions 1 and 2 are close to the spray nozzle and the temperature changes greatly after the injection of liquid nitrogen. The fluctuations of temperature at monitoring positions 5 and 6 are small since they are far from the nozzle. With the increase in liquid nitrogen injection, the space is filled with liquid nitrogen, but the concentration of liquid nitrogen varies at different locations, resulting in different temperature variations of the heating sheets at different locations. It can be seen from Figure 4a,b that the 6510# and nozzles have the most significant cooling effect on the heating sheets which can be reduced by 35 °C. Although the diameter of the 6530# nozzle is largest, but the cooling effect is poor. It can be seen from Figure 4d that the minimum temperature on the side of the container decreases to about 12 °C within 600 s for the 6510# nozzle, the minimum temperature for the 6520# nozzle can drop to about 10 °C within 300 s, and the minimum temperature for the 6530# nozzle can drop to 16 °C. On the whole, the cooling effect of the 6520# nozzle is better than those of 6510# and 6530# nozzles.

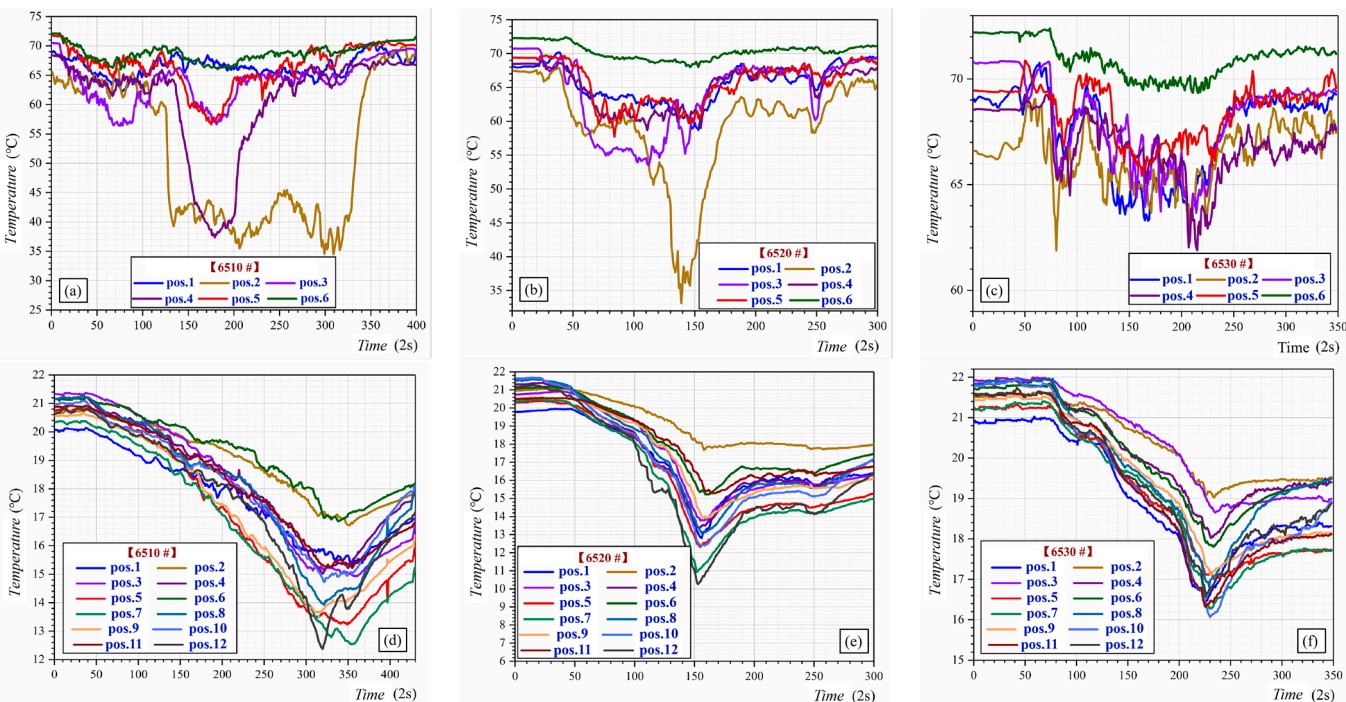

**Figure 4.** Temperature variations of different monitoring points after liquid nitrogen injection by 65 series nozzles: (**a**) Temperature variations on the heating sheets for 6510#; (**b**) Temperature variations on the heating sheets for 6520#; (**c**) Temperature variations on the heating sheets for 6530#; (**d**) Temperature variations on the side of the container for 6510#; (**e**) Temperature variations on the side of the container for 6520#; (**f**) Temperature variations on the side of the container for 6530#.

As shown in Figure 5, the temperature at the monitoring positions 2 and 4 of the heating sheets on the boxes fluctuates greatly and drops in a very short time when the 11010# nozzle is used, indicating the significant cooling effect. When 11020# nozzle is used, the temperature at position 5 fluctuates obviously, and the lowest temperature reaches 32 °C. For the monitoring points on the side of container, the temperature change at position 7 is the largest, and at position 2 is the smallest when 11010# nozzle is used, with the lowest temperature reaching 12.2 °C. When using the 11020# nozzle, the temperature changes little in the initial stage, but the temperature of each position of drops to the lowest point in a short time. In comparison, it can be seen that the maximum temperature drop of each monitoring point for 11010# nozzle is larger than that of 11020# nozzle.

The liquid nitrogen sprayed by the quarter sector solid conical spray nozzles 5B# and 10B# will form a cone eddy spray area due to the action of gravity and centrifugal force. If the injection pressure in this area is less than 0.2 MPa, continuous round jets may be generated at the spray outlet, thus forming a large number of large-diameter droplets [29], as shown in Figure 6a. By comparing Tables 3 and 4, it can be found that the temperature of the heating sheets at positions 1 and 3 fluctuates greatly when the 5B# nozzle is used, and the temperature of the heating sheets at positions 2 and 5 changes obviously when the type 10B# nozzle is used. This is because the solid conical spray nozzles is easy to form the frosting phenomenon, plugging the outlet of liquid nitrogen nozzles, as shown in Figure 6b. For the monitoring points on the side of container, it can be seen from Tables 3 and 4 that the minimum temperature at the monitoring points can decline to 14.6 °C when the 5B# nozzle is used. However, the temperature of each monitoring point has little change when the 10B# nozzle is used. Overall, in terms of temperature drop, the maximum temperature drop for the 5B# nozzle is greater than that of the 10B# nozzle.

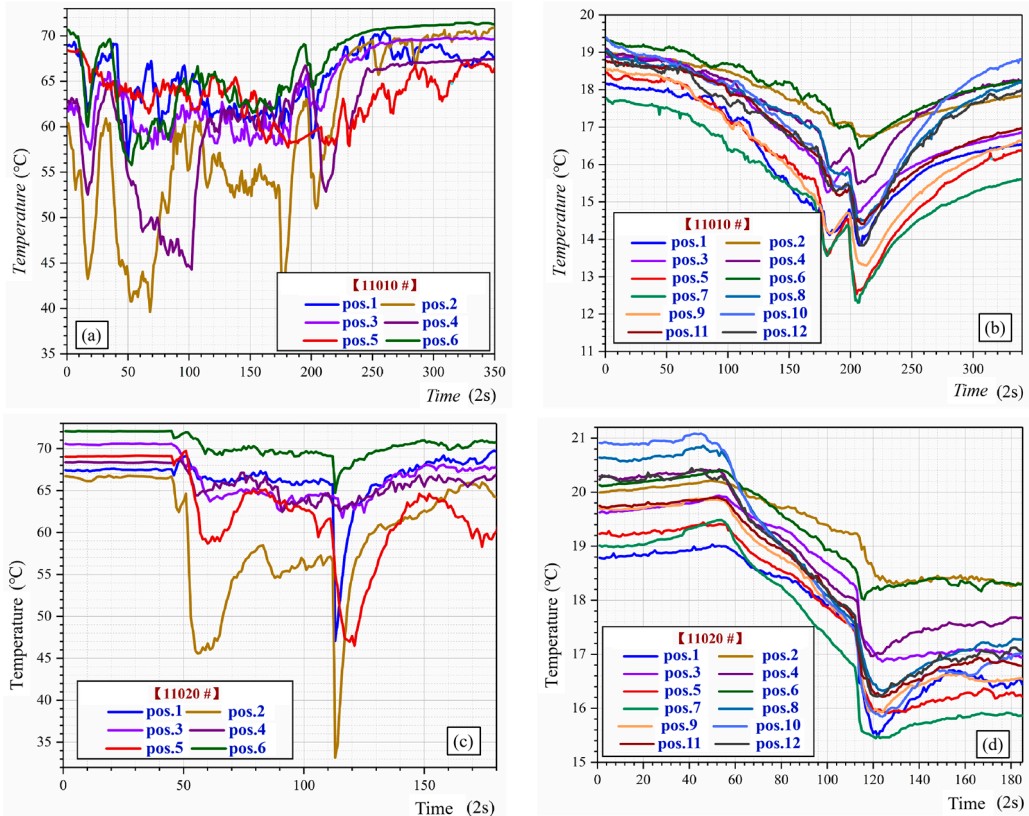

**Figure 5.** Temperature variation of different monitoring points after liquid nitrogen injection by 110 series nozzles: (**a**) Temperature variations on the heating sheets for 11010#; (**b**) Temperature variations on the side of the container for 11010#; (**c**) Temperature variations on the heating sheets for 11020#; (**d**) Temperature variations on the side of the container for 11020#.

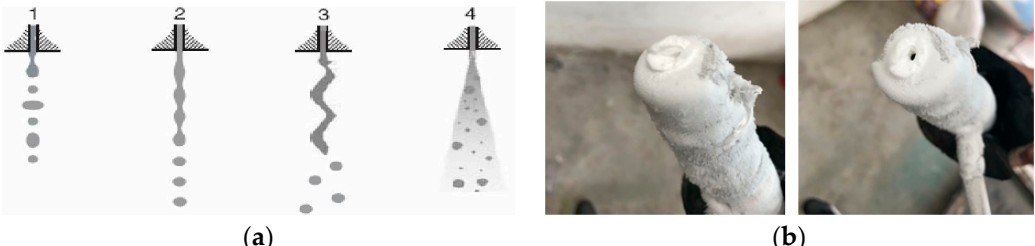

**Figure 6.** Liquid nitrogen ejection from solid conical spray nozzles: (**a**) Different patterns for atomization of solid nozzles; (**b**) Frosting phenomenon at the outlet of nozzles.

**Table 3.** Temperature change of heating sheet for the quarter sector solid conical spray nozzles.

| Nozzle | Temperature * | Monitoring Position | | | | | |
|--------|---------------|---|---|---|---|---|---|
| | | 1 | 2 | 3 | 4 | 5 | 6 |
| 5B# | $T_0$ | 1.0 | 2.3 | 3.2 | 3.9 | 4.6 | 5.1 |
| | $T_{min}$ | 1.9 | 4.6 | 6.5 | 7.9 | 9.1 | 10.2 |
| 10B# | $T_0$ | 1.0 | 2.3 | 3.2 | 3.9 | 4.6 | 5.1 |
| | $T_{min}$ | 1.9 | 4.6 | 6.5 | 7.9 | 9.1 | 10.2 |

* $T_0$ represents the initial temperature and $T_{min}$ represents the minimum temperature, °C.

**Table 4.** Temperature change on the test bench for the quarter sector solid conical spray nozzles.

| Type | Temperature * | Monitoring Position | | | | | | | | | | | |
|------|---------------|------|------|------|------|------|------|------|------|------|------|------|------|
| | | 1 | 2 | 3 | 4 | 5 | 6 | 7 | 8 | 9 | 10 | 11 | 12 |
| 5B# | $T_0$ | 18.3 | 19.5 | 19.4 | 19.7 | 18.9 | 19.5 | 18.1 | 19.7 | 19.0 | 20.2 | 19.0 | 19.6 |
| | $T_{min}$ | 14.8 | 18.1 | 16.9 | 17.8 | 15.3 | 18.5 | 15.3 | 18.0 | 14.9 | 17.1 | 14.6 | 17.5 |
| 10B# | $T_0$ | 17.9 | 19.2 | 19.0 | 19.6 | 18.5 | 19.5 | 17.8 | 19.5 | 18.4 | 19.9 | 18.4 | 19.5 |
| | $T_{min}$ | 16.7 | 18.3 | 17.2 | 17.8 | 16.0 | 18.4 | 15.8 | 17.4 | 16.8 | 17.6 | 17.3 | 16.9 |

\* $T_0$ represents the initial temperature and $T_{min}$ represents the minimum temperature, °C.

### 3.2. The Variations of Oxygen Concentration in Cooling Process

When the liquid nitrogen is injected, the oxygen concentration in the test container decreases obviously, but the use of different nozzles leads to different results. The overall change trend of oxygen concentration at different monitoring points is similar. Taking monitoring points 6 and 7 as examples, Figure 7 shows the change process of oxygen concentration with time when different types of nozzles are used. When using the 65 series quarter sector fan-shaped nozzles, the oxygen concentration drops from about 21% to below 10% and eventually tends to be stable. For the 6510# and 6530# nozzles, the oxygen concentration remains stable in the initial stage and gradually decreases with the increase of liquid nitrogen injection. For the 6520# nozzle, the oxygen concentration drops rapidly within a short time of liquid nitrogen injection. For the 11020# nozzle, the oxygen concentration decreases from the initial 20.5 to 10% in a short time, and for the 11010# nozzle, the oxygen concentration decreases from 21.5 to 12%. When using the 5B# and 10B# nozzles, the oxygen concentration decreases from about 21% to about 11%.

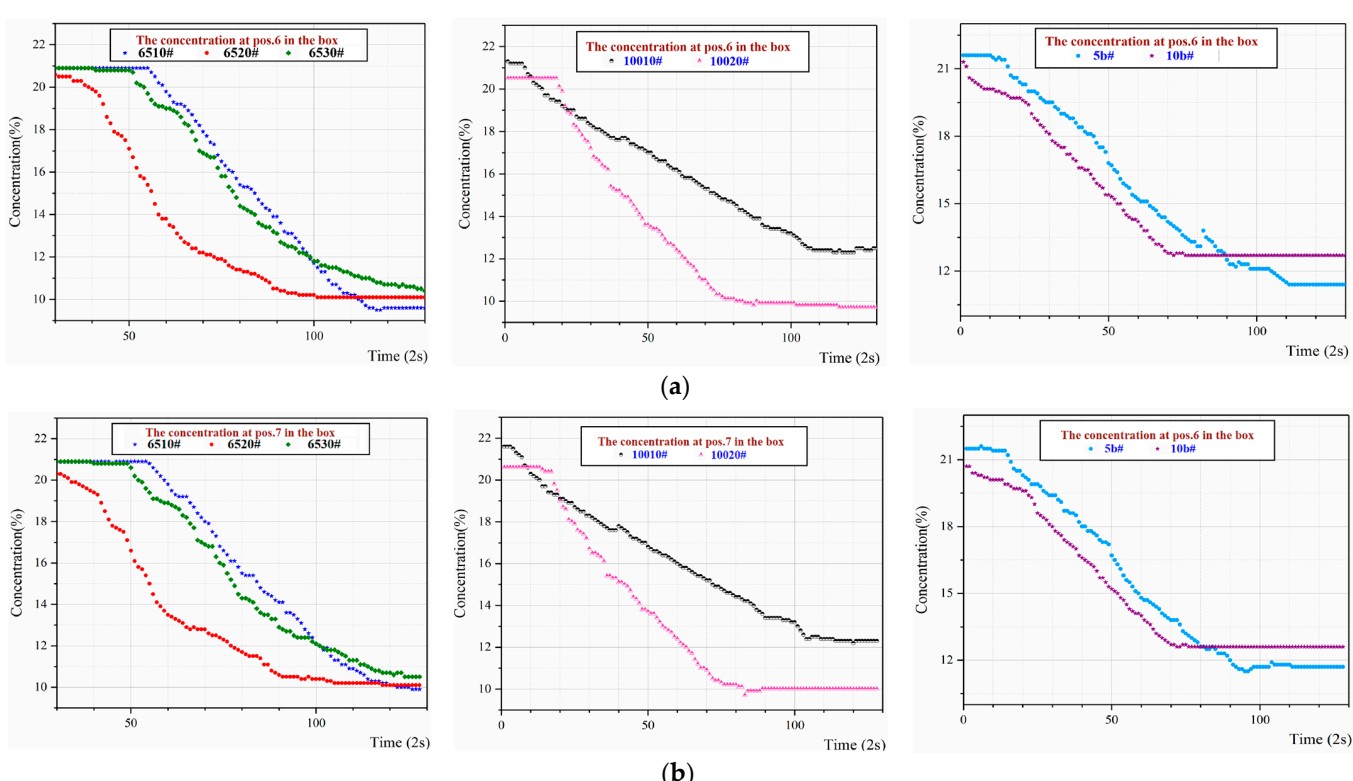

**Figure 7.** Variations of Oxygen concentration in the test container for different nozzles: (**a**) Variations of oxygen concentration with time at position 6; (**b**) Variations of oxygen concentration with time at position 7.

Moreover, the comparison of Figure 7a,b shows that for the same monitoring points, the difference in the final oxygen concentration is smallest when using the 65 series of nozzles, and the difference is biggest when using the 110 series nozzles.

Combined with the variations of the heating sheet temperature, the test bench temperature, and the oxygen concentration in the space, it can be seen that the temperature in the computer data center room is significantly reduced when injecting liquid nitrogen with the 6520# nozzle, and the temperature of the heating sheet and the test bench are reduced to 35 °C and 10 °C, respectively. In addition, the oxygen concentration under this condition decreases the fastest in a short time compared with other types of nozzles. To sum up, the cooling effect and suffocation effect on the space are the best when the liquid nitrogen is injected with the 6520# nozzle.

### 3.3. Extinguishing Effects of Liquid Nitrogen on Optical Fiber Combustion

In the simulation, the 6520# nozzle is used to test the extinguishing effects of liquid nitrogen on optical fiber combustion. Due to the flame retardant of the surface material of optical fibers, it is found that the flames die out in a short time after the optical fibers are ignited, releasing white smoke, but the temperature at the ignition point is still very high. In a fire in the data center room, optical fibers usually burn along with other combustible materials. Therefore, the optical fibers are wrapped in an integrated fiber tray to burn, as shown in Figure 8. Then, the fire extinguishing test with liquid nitrogen is carried out.

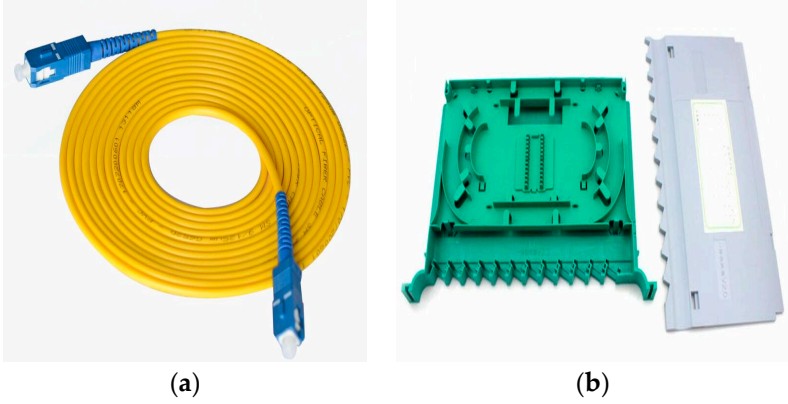

(**a**)　　　　　　　　　　　　　　　　　　　　(**b**)

**Figure 8.** Combustible materials used in the optical fiber combustion test: (**a**) Optical fiber; (**b**) Integrated fiber tray.

At the initial stage of combustion, the optical fiber burns in a small fire which is stabilized after burning for a period of time. Then, the liquid nitrogen is injected into the test container at the flow rate of 0.8 kg/s. The temperature and oxygen concentration in the container is steadily reduced, and the optical fiber and wrapped combustible materials are gradually extinguished due to the lack of enough oxygen. Figure 9 shows the ignition point spread and flame extinguishing process in the fiber burning process. Finally, it can be seen that although the ignition point has a tendency to extend outward, it can be quickly controlled. The combustion flame is completely extinguished within 3 min after injection of liquid nitrogen.

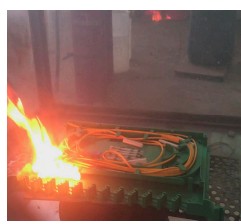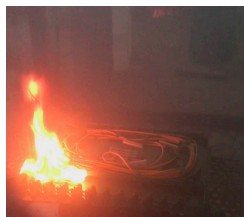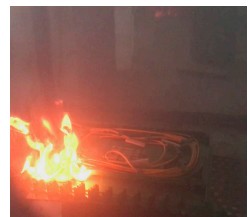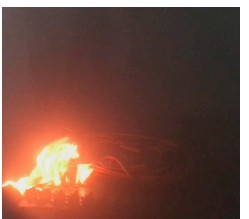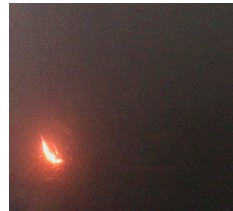

**Figure 9.** Optical fiber fire extinguishing scene.

### 3.4. Extinguishing Effects of Liquid Nitrogen on Combined Combustion of UPS

The main combustible materials of uninterruptible power supply (UPS) are battery electrolytic solution, circuit board and its surface combustible materials. The combination of the UPS circuit board and switch transformer are used in the test, as shown in Figure 10. The test process of UPS combustion is similar to that of optical fiber combustion. It can be seen in Figure 11 that the whole combustion process presents a small fire. Compared with optical fiber combustion, the ignition point range of the uninterruptible power supply combination fire is smaller. Although liquid nitrogen is injected at a low flow rate, the fire is successfully extinguished within 2 min. It indicates that liquid nitrogen has obvious extinguishing effect on the combustion of uninterruptible power supply combination.

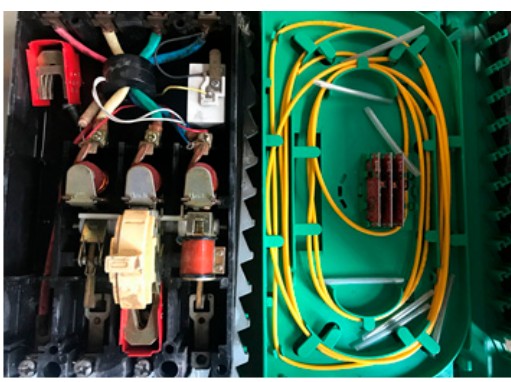

**Figure 10.** Uninterruptible power supply combination.

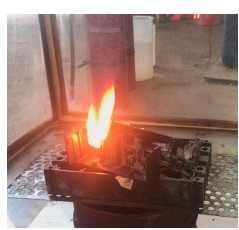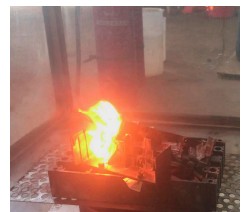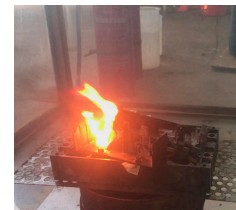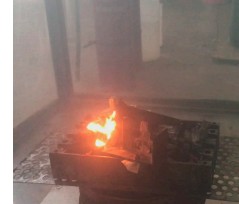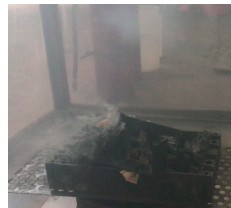

**Figure 11.** UPS fire extinguishing scene.

### 4. Conclusions

Although liquid nitrogen has been widely used in the fire suppression in coal mines and other industries, there is little research on its application in the computer data center room. This study preliminarily simulates the temperature and oxygen concentration variations in a data center room when liquid nitrogen was injected using seven kinds of nozzles, and the fire-extinguishing effect of liquid nitrogen on two combustible materials. The main conclusions are as follows:

(1) At the same nitrogen injection pressure, the largest peak temperature drop in the computer data center room is observed when nitrogen is injected using the 6520# nozzle among the seven nozzles. The temperature of the heating sheet and the test bench are reduced to 35 °C and 10 °C, respectively.

(2) When nitrogen is injected using different types of nozzles, the oxygen concentration at the measurement points in the computer data center room exhibits a similar variation tendency, all remaining constant at first, then decreasing rapidly, and finally stabilizing. Compared to other types of nozzles, oxygen concentration drops fastest in a short time under the 6520# nozzle.

(3) Compared with optical fiber, the ignition range of uninterruptible power supply combinations during combustion is relatively small. Liquid nitrogen has significant fire-extinguishing effect on two combustible materials, which can successfully extinguish optical fiber and UPS within 3 min and 2 min, respectively.

**Author Contributions:** Conceptualization, J.M. and J.K.; methodology, G.L.; software, T.W.; validation, J.M. and T.W.; formal analysis, J.M.; investigation, G.L.; resources, J.M.; data curation, T.W.; writing—original draft preparation, J.M.; writing—review and editing, T.W.; visualization, J.M; supervision, J.K.; project administration, J.M; funding acquisition, J.K. All authors have read and agreed to the published version of the manuscript.

**Funding:** This research was funded by the Qinglan Project of Jiangsu Province.

**Institutional Review Board Statement:** Not applicable.

**Informed Consent Statement:** Not applicable.

**Data Availability Statement:** Not applicable.

**Conflicts of Interest:** The authors declare no conflict of interest.

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
