# Peer review of "Simulation Test on Cooling and Fire Suppression with Liquid Nitrogen in Computer Room of Data Center"

_fire, doi:10.3390/fire6030116_

Round 1
Reviewer 1 Report
In this paper, the liquid nitrogen cooling process in the initial heating stage of computer data center room fire and the liquid nitrogen extinguishing effects for common combustible materials were studied by using the self-designed simulation platform of liquid nitrogen spray. Overall, the research work has scientific soundness. The manuscript can be accepted for publication after addressing the following comments:
1 The important information on experimental conditions should be supplemented, such as initial ambient temperature and ambient pressure.
2 The “Type” of the spray nozzle in Table 1 and Table 2 should be added with “#”, such as 6510#. And the serial number of Table 1 is incorrectly marked.
3 The title of Figure 11 and Table 4 seem to be incorrect and need to be modified.
4 There are grammatical and spelling errors in some sentences. They should be modified.
(a) Therefore, the afe and reliable fire protection system is very important for the stable operation of data center room.
(b) Additional equipment such as infrared thermal imager and liquefied butane gas cylinder (igniter) are needed,
(c) Fire hazards in data center mainly include electirc cable,
(d) A total of 12 4OXV oxygen sensors and 18 K-type thermocouples are palced inside the test container
(e) It can be seen from Figure 4 (d) that the minimum temperature on the side of container dencreases to about 12℃
(f) the 6520# nozzle are used to test the extinguishing effects of liquid nitrogen
(g) The main combustible materials of uninterruptible power supply (UPS) are battery electrolylic solution,
(h) This study preliminarily simulates the temperature and oxgen concentration variations
Author Response
We really appreciate for your patience and comments, which help us greatly promote our paper. According to your comments, the manuscript has been carefully revised point by point, and all revisions were marked in red in the revised manuscript. The response to each comment is summarized in the following file.

Reviewer 2 Report
This paper presents a detailed study on the cooling and inert effects of liquid nitrogen on the data center room fire at different nozzle flows, as well as the fire-extinguishing effects of liquid nitrogen on two combustible materials. Related works are useful for guiding practical application. There are several minor issues that need to be solved.
(1) The dimension of the six boxes used to simulate the computers in the data center room should be given.
(2) In section 3.2, “Moreover, the comparison of Subfigures 8 (a) and (b) shows that for the same monitoring points.” There should be Subfigures 7 (a) and (b) not Subfigures 8 (a) and (b).
(3) There are some writing errors in this paper, such as “draw out the stainless steel pipe, , and replace different types of nozzles”; “as shown in Figure 1(a).The 4OXV oxygen sensors and heating sheets”, “Through repeated attempts and and considering various influencing factors”, which need to be corrected.
(4) It is recommended to adjust Figure 8 to Section 3.3.
Author Response

(The authors gave the same response as above.)

Reviewer 3 Report
This work presents some experiments of liquid nitrogen discharge within a compartment that should reproduce a computer room. The subject can be appealing to the readership of Fire; however, substantial action is advised towards a general improvement of the contents, which also required additional analysis.
Main:
1. The most critical concern about your study consists of the lack of analysis that would substantiate your observations. Basically, you released a certain amount of nitrogen within a compartment upon reaching a threshold temperature in some solid bodies (i.e., boxes) placed within, then a set of combustibles (e.g., cables) was ignited prior to another release of nitrogen. You claim that a certain nozzle (no. 6520) “has a great influence on the temperature in the container which can be reduced to 10℃ in a short time” and its “cooling effect” “is the best among seven tested nozzles”. Why? Nitrogen combines suffocation and cooling towards fire suppression: you should relate oxygen depletion and heat removal to flammability of the tested materials.
2. Quite unexpectedly, operative pressure is not reported as a quantitative information (“Under a certain pressure”, “If the injection pressure in this area is insufficient”). Actually, that is crucial in whatever discharge system and necessary to identify the discharge conditions in Tables 1 and 2. I am also wondering whether the comparison between nozzles was made by keeping pressure or flow rate constant, or none of the two.
3. Much related to the previous point, it is unclear why nozzle no. 6520 exhibited the best performance: it appears you also tested a nozzle discharging larger flow rate at the same operative pressure (i.e., no. 6530), the action of which should have led to suppression before that of nozzle no. 6520. In fact, one of the main objectives of fire safety engineering is assessing a threshold beyond which suppression is not achieved. However, nothing about that is mentioned in your work at all, since no failed suppression is reported by any of the tested nozzles. To this end, you may expand your analysis as inspired by past research on other fire protection systems (e.g., P.E. Santangelo et al., Fire Safety Journal 70, 98-111, 2014; J. Shi et al., Geomatics, Natural Hazards and Risk 11, 22-39, 2020), which you may also consider including in your reference list.
4. As hinted at in the previous point, it seems the compartment used in your experiments had such a small volume that suppression could be reached in all the tested conditions. However, the most significant contribution would consist of scaling the tested systems up to the actual size of data centers, which would basically imply scaling the discharge rate by some sort of characteristic length of the compartment, likely its height (J.G. Quintiere, Fire Safety Journal 15, 3-29, 1989). You may suggest some approach to translate your observations into some general guidance for nitrogen-based fire protection systems.
5. The entire analysis performed on thermograms is very questionable, especially since you seem to apply infrared thermography to detect temperature quantitatively in a fire scenario. Unfortunately, there are two main factors against that: emissivity of materials undergoing a combustion reaction is quite unknown, so I am not sure about the image processing – completely missing from the reported information – you applied to your dataset; fire implies radiation mostly in the visible light spectrum, which does not overlap with infrared. I am seriously concerned about the contents included in Sections 3.3 and 3.4; actually, the temperature values shown in both contour and trend plots (Figs. 9b, 9c, 10, 12b, 12c and 13) seem completely unsubstantiated and arguably wrong.
6. No statistical analysis is proposed about your dataset at all: accuracy and repeatability should be evaluated and reported, together with clarifying the number of repeats conducted for each tested configuration.
7. The introduction presents a mere account of some previous works, but how those studies and the reported findings were useful to your research is quite unclear. Moreover, you may acknowledge and cite the relatively few works available on fire in data centers and electronic equipment (e.g., B.J. Meacham, Fire Technology 29, 34-59, 1993; M. Strianese et al., Applied Sciences 13, 186, 2023).
8. The conclusions may be revised and probably rewritten in the light of the previous comments and additional analysis requested. As a suggestion, very generic statements (e.g., “The main factors affecting the fire extinguishing effects are the characteristics of the material itself and the type of nozzle.”) should be substantiated by quantitative results and some deeper discussion.
Additional comments are listed below.
Section 2. The employed thermocamera should be described (i.e., model, manufacturer, sensor, resolution) and acquisition frequency may be reported.
Section 2. “Equivalent aperture” arguably means characteristic orifice diameter, equivalent orifice diameter or equivalent diameter.
Section 3. The legend “uninterruptible power”, probably hinting at UPS, is completely unrelated to the dataset presented in Fig. 13.
Some typos (e.g., “palced” instead of “placed”) and the need for spelling acronyms out the first time they appear (e.g., UPS, DT, IT) suggest a thorough revision of the language.
Author Response

(The authors gave the same response as above.)

Round 2
Reviewer 3 Report
No further comments.